# Rice Husk—Cellulose-Based Agricultural Waste Enhances the Degradation of Synthetic Dyes Using Multiple Enzyme-Producing Extremophiles

**DOI:** 10.3390/microorganisms11081974

**Published:** 2023-07-31

**Authors:** Van Hong Thi Pham, Jaisoo Kim, Soonwoong Chang, Jeahong Shim, Woojin Chung, Donggyu Bang

**Affiliations:** 1Department of Environmental Energy Engineering, College of Creative Engineering, Kyonggi University, Suwon 16227, Republic of Korea; 2Department of Life Science, College of Natural Science, Kyonggi University, Suwon 16227, Republic of Korea; jkimtamu@kyonggi.ac.kr; 3Soil and Fertilizer Management Division, National Institute of Agricultural Sciences, Rural Development Administration, Wanju-Gun 55365, Republic of Korea; jaysoil@korea.kr; 4Department of Environmental Energy Engineering, Graduate School, Kyonggi University, Suwon 16227, Republic of Korea; ahr1emd@kyonggi.ac.kr

**Keywords:** synthetic dyes, microbial dye degradation, extremophiles, rice husks

## Abstract

The brightly colored synthetic dyes used in the textile industry are discharged at high concentrations—for example, various azo dyes including Methylene Blue (MB) and Methyl Orange (MO)—which is a matter of global concern, as such dyes are harmful to humans and the environment. Microbial degradation is considered an efficient alternative for overcoming the disadvantages of conventional physical and chemical dye removal methods. In this study, we investigated the potential of multiple types of the enzyme-producing extremophilic bacteria *Bacillus* FW2, isolated from food waste leachate, for the decolorization and bioremediation of artificial synthetic dyes. The screening of enzyme production and assaying of bacterial strain enzymes are essential for enhancing the breakdown of azo bonds in textile azo dyes. The degradation efficiencies of the water-soluble dyes MB and MO were determined at different concentrations using rice husk, which is an efficient substrate. Using the rice husks, the MO was removed completely within 20 h, and an estimated 99.8% of MB was degraded after 24 h by employing shaking at 120 rpm at 40 °C—whereas a removal efficiency of 98.9% was achieved for the combination of MB + MO. These results indicate the possibility of applying an extremophilic bacterial strain, *Bacillus* sp., for large-scale dye degradation in the future.

## 1. Introduction

The massive discharge of azo dye industrial effluents is an emerging concern, owing to their damaging effects on humans and on the environment [1]. Textile industries use a wide range of highly toxic chemicals, including toxic dyes; heavy metals such as mercury, chromium, cadmium, lead, and arsenic; and additives during the process of color pigment production. However, these substances do not adhere tightly to the fabrics and, therefore, are easily released into aquatic environments along with wastewater [2,3]. Consequently, these substances pose serious risks to terrestrial growth by changing soil properties and fertility, and affect aquatic life by reducing light penetration and increasing the biological oxygen demand (BOD), chemical oxygen demand (COD), salinity, and alkalinity—which inhibit the growth of microorganisms [2,4]. Excess dye effluents released into the environment enter the food chain, causing recalcitrance, biosorption, and bioaccumulation in the human body—which potentially promote toxicity, mutagenicity, and carcinogenicity [5,6]. Therefore, it is necessary to treat dye-contaminated wastewater before it is discharged into the environment. Several approaches have contributed significantly to dye degradation, including physicochemical, nanoparticle-based, and biological methods [7,8]. However, biological treatment strategies are the most appropriate because of their eco-friendliness, economic feasibility, and less-toxic secondary production. Bioremediation using microorganisms is a valuable biological method for the decolorization of various azo dyes. Among such treatments, bacteria have received more attention because of their faster growth and higher dye degradation efficiency compared with that of other microbes [9,10]. Bacteria adapt well to various types of stresses, including hazardous wastes and harsh living conditions, which are considered positive features for the enhancement of many contaminated environments [11]. In particular, the activity of bacterial enzyme systems contributes to the biotransformation of toxicants into non-toxic metabolites through metabolic mechanisms [12]. Several types of enzymes have reportedly been found to be effective molecular weapons for participating in dye degradation, including oxidative enzymes (polyphenol oxidase (PPO), manganese peroxidase (MnP), lignin peroxidase (LiP), laccase (Lac), tyrosinase (Tyr), and N-demethylase), reductive enzymes (azoreductases), and immobilized enzymes (oxidoreductases)) [13,14].

Numerous bacterial strains have been investigated as potential candidates for functional bioremediation agents to decolorize and mineralize dyes that contaminate soil and wastewater. In a previous study, a bacterium *Paenibacillus terrigena* KKW2-005—isolated from dye-contaminated soil—was identified and its ability was determined to be capable of decolorizing 96.45% of the azo dye Reactive Red 141 (50 mg/L) within 20 h, at pH 8.0 and a wide range of temperatures (30–40 °C) under static conditions [12]. Pham et al. (2022) investigated a novel member of the genus *Bacillus*—namely, *Bacillus* React3—which can degrade Methylene Blue (MB) during the production of the lignin peroxidase enzyme [11]. A novel aerobic bacterial strain—*Bacillus cereus* ROC—degrades azo dyes (Reactive Orange 16 and Reactive Black 5) under aerobic and agitation conditions, with both free and immobilized cells, within 5 d [15]. A recent report indicated that a consortium of three haloalkaliphilic isolates belonging to the genus *Halomonas* showed a remarkable ability to decolorize Reactive Black 5 (87%) [16]. Another alkaliphilic, halotolerant bacterial strain EMLA3 showed 97% degradation of 50 mg/L Methyl Red after 16 h at an initial pH of 11.5 [17]. Pandey et al. found that *Bacillus*. sp. decolorized 92% of the free cells of Reactive Orange 16, as well as 97% of its immobilized cells. The free cells of *Lysinibacillus* sp. removed 95% of the Reactive Blue 250 dye from the immobilized cells at 99% efficiency [18].

Previous studies have shown that rice husks, a cellulose-based agricultural waste, act as an efficient adsorbent material for dye-contaminated wastewater [10,19]. Based on the reported chemical composition of rice husks of 32.2% cellulose, 21.3% hemicellulose, 21.4% lignin, 1.8% extractives, 8.1% water, and 15.1% SiO_2_, rice husks are natural cellulosic fibers that have strong potential for dye adsorption and decolorization [20,21]. However, few studies have focused on the function of rice husks as a sustainable carbon source for microbial activity. Forss and co-workers (2017) found that rice husk biofilters can support a microbial community by more than 90% over 67 h to enhance the biodegradation of textile wastewater [22]. In another study, rice husks were found as a significant substrate for growing microorganisms and for the efficient removal of organic matter and nitrogen compounds in wastewater treatments [23]. Accordingly, this study aimed to investigate multifunctional extremophilic bacterial candidates that can utilize agricultural waste such as rice husks as carbon sources—thereby accelerating the removal of dye-polluted effluents. Additionally, using raw materials without pretreatment will minimize the cost of the treatment process. Moreover, the isolate used in this study showed great ability for multiple enzyme production, which could contribute to the highly significant efficiency of dye degradation via enzymatic pathways and accumulation. Using such a method would prevent the need for the secondary treatment of dye-adsorbed rice husks—which may cost more and unpredictably bring other, consequent environmental problems—if it contributed to decolorization as a biosorbent.

## 2. Materials and Methods

### 2.1. Sampling and Isolation of the Degrading Bacteria of Azo Dyes

Following a previous study, food waste was collected from the Jowon Industry in South Korea and used as an isolation source for the target dye-degrading bacteria. The selective media included skim milk, starch, and carboxymethylcellulose (CMC) as carbon and nitrogen sources; a vitamin solution; and autoclaved soil extract to provide vitamins and minerals. All samples were tested at different temperatures from low to high levels (–6 °C–80 °C), and under a wide range of pH levels (4–12.5) for 5 d. The active bacterial candidates were selected for further experiments [8,24].

### 2.2. Identification and Phylogenetic Analysis

Genomic DNA was extracted from the bacterial strain according to the manufacturer’s instructions using an InstaGene Matrix kit (Bio-Rad, Hercules, CA, USA) for the polymerase chain reaction (PCR) amplification of the 16S rRNA gene. A pair of universal primers, 27F and 1492R, was used to amplify the 16S rRNA gene [25]. The PCR products were purified using a multiscreen filter plate (Millipore Corp, Bedford, MA, USA) and sequenced using the primers 518F (5′-CCA GCA GCC GCG GTA ATA CG-3′) and 800R (5′-TAC CAG GGT ATC TAA TCC-3′), with a PRISM BigDye Terminator v3.1 Cycle Sequencing Kit (Applied Biosystems, Foster City, CA, USA). The nearly full-length 16S rRNA sequence was assembled using SeqMan software (DNASTAR Inc., Madison, WI, USA). Sequence similarity was determined by comparison with existing sequences available in the GenBank database using the EZBioCloud server [26]. The MEGA 7 program was used to align sequences and reconstruct phylogenetic trees based on the BLAST results [25,27].

### 2.3. Determination of Ligninolytic Enzyme Production of Strain FW2

Ligninolytic enzymes, including laccase (Lac) and lignin peroxidase (LiP), were assayed and analyzed spectrophotometrically in cell-free extracts.

The Lac activity was determined through the oxidation of 2,2′-azinobis-(3-ethylbenzethiazoline-6-sulphonate; ABTS), following the method of Wolfenden and Wilson [24]. A 0.15 mL enzymatic mixture of 0.03% ABTS, 0.5 mL of 0.1 M sodium acetate buffer at pH 5.0, and 0.35 mL of supernatant of bacterial culture was used. The formation of oxidized ABTS was measured in a 1 mL quartz cuvette using a UV spectrophotometer at a wavelength of 530 nm. The oxidation of ABTS was followed by an increase in absorbance at 420 nm [28].

For the LiP enzyme assay, LiP activity was determined by the oxidation of veratryl alcohol to veratryl alcohol at 310 nm. One milliliter of enzymatic mixture in assay consisted of 0.4 mL of citrate-phosphate buffer (100 mM, pH 2.7), 0.1 mL of veratryl alcohol (20 mM), and 0.5 mL of fluid sample. A daily addition of 40 µL of H_2_O_2_ (20 mM) was employed to initiate the reaction. Conversion to veratraldehyde was monitored in a 1 mL quartz cuvette using a UV spectrophotometer at a wavelength of 310 nm. One unit of enzyme activity corresponded to the oxidation of 1 µM veratraldehyde, converted from veratryl alcohol, per minute under the assay conditions [29].

### 2.4. Degradation and Decolorization Experiments of Dyes

Before these experiments, bacterial cells were prepared under optimal growth conditions (pH 7; 40 °C), and in an optimal medium. Experiments were performed using different concentrations of bacterial cells (10^3^–10^8^ CFU/mL, interval ×10^1^ CFU/mL), MB, and Methyl Orange (MO; 0–250 mg/L, interval 50 mg/L). All experiments were conducted in triplicate. The concentration of each dye was adjusted to 100 ppm in the LB medium for the degradation of mixed dyes.

Samples were collected every 24 h to determine their decolorization and degradation capacities. Bacterial cultures were centrifuged to remove biomass, and the supernatant containing the residual dye was assessed to determine degradation by the change in absorbance of clear supernatants at an absorption λ_max_ of 665 nm and 465 nm, using an ultraviolet–visible (UV–Vis) spectrophotometer for the MB and MO, respectively.
$$\mathrm{MB}/\mathrm{MO}\;\mathrm{degradation}=\frac{\mathrm{Innitial}\;\mathrm{absorbance}-\mathrm{Absorbance}\;\mathrm{after}\;\mathrm{degradation}}{\mathrm{Initial}\;\mathrm{absorbance}}\;\times \;100$$

### 2.5. Effects of Physicochemical Factors on Bacterial Growth and Dye Degradation

Different growth parameters including the temperature, pH, NaCl, and carbon and nitrogen sources were used by in an experiment conducted by Pham et al. (2022) [11]. The pH gradients were set up ranging from 4 to 12 (1 unit intervals), with different buffers (PBS buffer (pH 4.0–7.0) and Tris–HCl buffer solution (pH 7.0–12.0)), with the temperatures set to 10 °C, 25 °C, 35 °C, 40 °C, 45 °C, 50 °C, 55 °C, and 60 °C. The carbon sources containing glucose, sucrose, dextrose, cellulose, and nitrogen were yeast extract, tryptone, peptone, and urea, respectively (Oxoid-UK, part of ThermoFisher Scientific, UK). The NaCl concentration ranged from 0% to 30% in the Luria–Bertani (LB) growth medium (intervals of 1%). The data were recorded at 12 h intervals over a period of 0 h to 72 h.

### 2.6. Effects of Rice Husks as a Special Carbon Source for FW2 Growth and Dye Degradation

The rice husks collected from an agricultural area in Cheolwon-gun, Gangwon-do used in this study are raw materials without any pretreatment. The bacterial culture medium contained (g/mL) yeast (0.5), blended rice husks (0.5), NaCl (1%, *w*/*v*), and 100 mg of individual and mixed dyes (50 mg for each). Two controls—with and without the bacterial inoculum—and another two controls, with and without the dye, were used for comparison. The culture was incubated at 40 °C, 120 rpm for 5 d.

### 2.7. Effects of Metal Ions on Enzyme Production and Dye Removal

To determine the effect of metal ions on 100 ppm-dye decolorization, the reaction solution was mixed with different metal ions (FeCl_2_, FeCl_3_, CaCl_2_, MnCl_2_, NiCl_2_, MgCl_2_, CoCl_2_, CuSO_4_, ZnCl_2_, and CdCl_2_) at a concentration of 2 mM, in addition to bacterial inoculums at 10^8^ CFU/mL. The residual activity of the mixtures was determined for decoloring the dyes. Three replicates were performed for each experiment.

### 2.8. Effect of Different Dye Concentrations

A concentration series of each dye (500, 800, 1000, 2000, and 3000 mg/L) was prepared to monitor the degradation/decolorization by enzymes produced from the strain FW2—after which, the concentration was determined using a spectrophometer at an λ_max_ of 665 nm and 465 nm for the MB and MO, respectively.

## 3. Results

### 3.1. 16S rRNA Gene Analysis

The bacterial strain FW2 showed the highest pairwise similarity with *Bacillus amyloliquefaciens* DSM 7^T^ (99.86%), based on a complete 16S rRNA sequence [24].

### 3.2. Optimal Conditions for Bacterial Growth, Enzyme Production, and Dye Degradation

#### 3.2.1. Effect of pH, Temperature, and NaCl

Strain FW2 is known as an extremophile as it can grow under a wide range of temperatures (−6 °C; 75 °C), pH levels (4.5; 12), and NaCl concentrations (up to 35%), and as it produces multiple enzymes such as amylase, protease, cellulase, and lignin peroxidase. The optimal physiochemical conditions for the growth and enzyme production of strain FW2 were observed at 40–45 °C, pH 7–8, and NaCl 0–15% [7,20]. The dye degradation efficiency peaked at almost 100% at pH 7.5, 40 °C, NaCl (0–15%), and 120 rpm after 6 h, but dyes were degraded almost completely after 24 h under static conditions (Figure 1a–c). The degradation rate decreased sharply at temperatures over 45 °C, pH 9, and 25% NaCl.

The effects of different temperatures were monitored to determine the optimal enzyme activity (Figure 2a). Dye degradation/decolorization by the lipolytic enzymes was produced by the bacterial strain within 6–40 h. The production of the LiP enzyme peaked at 40 U/mL, 40 °C, and pH 7.0, and the Lacs enzyme was generated at a maximum rate of 1.5 U/mL after 24 h in a shaking incubator at 120 rpm, 35 °C, and pH 7.0 (Figure 2b). This shows the correlation between enzyme activity and dye degradation efficiency, which increased along with increasing bacterial growth in the logarithmic phase.

The LiP increased the enzyme activity from 25 °C, reached its strongest activity level at 40 °C (40 U/mL), and then subsequently gradually decreased with increasing temperatures to 60% at 55 °C and 40% at 60 °C. The LiP showed strong activity over a wide range of temperatures, from 25 to 60 °C. The optimal temperature for the Lac activity was 35 °C with 1.5 U/mL, and it remained at 43% activity at 60 °C. The optimal pH for LiP and Lac activities was 7.0–7.5, respectively—gently decreasing from pH 8 and totally inhibited from pH 10, as shown in Figure 2b. The highest enzyme activity was observed under a wide range of NaCl concentrations—0–15%—and was reduced to 80% and 45% relative activity at 20% and 35% salt concentrations, respectively.

#### 3.2.2. Effect of Carbon and Nitrogen Sources

Figure 3a shows that the decolorization and degradation rates of MB and MO reached the highest percentage of nearly 100% in the presence of cellulose and glucose as the optimal carbon sources for bacterial growth and dye removal in the individual and mixed dyes. While glucose had the second-highest effect on the removal of the mixed dye, sucrose moderately contributed to the degradation efficiency. The presence of dextrose does not significantly contribute to dye decolorization rate. The maximum percentages of both the MB and MO, of almost 100% and 96% of the mixture, were observed with yeast extract as an optimal nitrogen source that may accelerate the whole process of the dye degradation. This was followed by the contribution of tryptone and peptone, which were recorded to have a moderate effect on dye removal—at 99% and 95%, respectively. The addition of urea did not much improve the dye decolorization.

### 3.3. Effect of Metal Ions

The metal ions Mn^2+^, Mg^2+^, Ca^2+^, Fe^2+^, Fe^3+^, Cu^2+^, Ni^2+^, Zn^2+^, Co^2+^, and Cd^2+^ were added to the assay to determine their dependence on enzyme activity and dye removal. Decolorization data for individual dyes and a mixture of MB and MO are shown in Figure 4. The presence of Cd^2+^ markedly inhibited the degradation degree to only 20 ± 1.6%, followed by Co^2+^, Ni^2+^, Fe^3+^, and Zn^2+^. In contrast, other metal ions—such as Ca^2+^, Fe^2+^, Mn^2+^, and Mg^2+^—accelerated the decolorization by 10%, 7%, 6%, and 4%, respectively, compared with the percentages achieved by the control samples (Figure 4).

### 3.4. Effect of Dye Concentrations on Bacterial Growth and Dye Removal Performance

Figure 5 shows that the effects of the initial dye concentration on the decolorization of FW2 bacterial growth decreased as the dye concentration increased. It is understandable that the dye molecules are interfering with the active sites of the functional dye-degrading enzymes, resulting in a decline of dye removal efficiencies under the operating conditions of this study. The minimum inhibitory concentrations of MB and MO were 2000 mg/mL and 3000 mg/mL, respectively (Figure 5).

Within 48 h, 100 ± 0.07%; 95 ± 0.05%; 90 ± 0.04%; 72 ± 0.06%; and 25 ± 0.05% of the mixed dyes were degraded in 500, 800, 1000, 2000, and 3000 mg/L dyes, respectively.

### 3.5. Effect of Rice Husk Concentrations on Bacterial Growth and Its Dye Degradation Activity

Figure 6 shows the bacterial growth line and dye removal capacity profile of FW2, as affected by different rice husk concentrations. Strain FW2 utilized rice husks as a carbon source for growth, with a peak at 5 g and a gradual decline with increasing concentrations.

## 4. Discussion

Different microorganisms exhibited optimal activity over a range of defined temperatures. The temperatures for efficient dye decolorization processes range from 20 to 37 °C [30,31]. However, some bacteria can degrade dyes at temperatures between 40 °C and 50 °C [32]. In Figure 1a, from 25 °C up to 40 °C, the relative activity of the enzymes increased linearly with bacterial growth and dye degradation. However, dye degradation declined when the enzyme production was reduced from 45 °C, whereas bacterial growth was not affected at this temperature. In a previous study, Chen et al. introduced a thermophilic *Anoxybacillus* sp. that could detoxify the azo dye Direct Black G at a rate of 98.39% of 100 mg/mL at 55 °C over 48 h [32]. Previous studies have shown that the optimum pH for dye decolorization is between 6.0 and 8.0, whereas the removal efficiency is low under acidic or alkaline conditions [33,34]. This study’s results indicated that a pH lower than six or higher than 7.5 inhibited enzyme production and dye removal, but not bacterial growth (Figure 1b). The bacterial strain FW2 was able to degrade both MB and MO over a wide range of salt concentrations (0–25% NaCl, *w*/*v*), but its efficiency was highest at a concentration range of 0–15% (Figure 1c). Salt is a significant factor in the treatment of dye wastewater and affects microbial growth and enzyme production. Generally, high concentrations of salt lead to low hydrophobicity, which is considered a factor that affects the plasmolysis of bacteria cells—causing a reduction in the growth of bacteria and a decrease in dye degradation [9]. Therefore, the salt tolerance of this bacterial strain plays a crucial role in dye degradation through intracellular accumulation inside the cell and absorption onto the cell surface.

Several extremophilic bacterial strains have been identified as potential candidates for enzymatic dye degradation. Thermophiles (optimally grown at 60 ± 80 °C) isolated from hot springs produce lignocellulolytic enzymes that are involved in dye degradation [35]. The thermophilic *Anoxybacillus* sp. PDR2 can remove the azo dye Direct Black G using different enzymes such as azoreductase and oxidoreductase [36]. Another study used a novel extracellular laccase isolated from *Geobacillus stearothermophilus* ATCC 10149, which could degrade up to 90% of Remazol Brilliant Blue R and other dyes such as MO, Malachite Green, and Indigo Carmine [37]. Similarly, in the current study, multiple enzyme-producing strains of FW2 were able to degrade various types of dyes at high temperatures through cell surface absorption, accumulation, and enzyme mechanisms. In particular, the stability of the enzyme activity of multifunctional extremophilic bacteria such as FW2 was higher than that of the strains tolerant to normal conditions, including the high adaptive response to oxidative stress that arises during dye degradation [38] (Figure 2).

As general carbon and nitrogen sources, strain FW2 is able to use a wide range of sources for its growth, and contributes significantly to the efficiency of dye degradation (Figure 3). This strain exhibited a high utilization of cellulose as the optimal carbon source for its growth, enzyme production, and dye removal. However, a previous study investigated starch as the best carbon source for the degradation of Red 2D dye using *Bacillus* sp. [39], while glucose was supplemented to enhance the degradation rate of the Azo dye basic Orange 2 by *Escherichia coli* [40]. Therefore, the significant utilization of cellulose as an optimal carbon source in the screening of enzyme production and dye removal clearly confirmed a crucial role for this extremophilic bacterial strain in dye degradation during the addition of rice husks.

The physio-biochemical properties and metabolism of microorganisms could be altered by the presence of toxic pollutants such as dyes containing heavy metals, which can cause oxidative damage and inhibit the bioremediation capacity of microbes. The solubility of heavy metals increases with increasing temperatures and rates of absorption, which improves the bioavailability of heavy metals in extremophilic bacterial strains. Therefore, investigating extremophiles that thrive under multiple environmental challenges and their potential for pollutant removal has recently become an attractive option for microbiologists. The Strain FW2 used in the current study showed a high tolerance for and was less affected by heavy metal stress, which was ascribed to the flexibility of its metabolic pathways under various conditions (Figure 4).

Strain FW2 can degrade dye at a concentration of 3000 mg/mL, which is higher than that achieved by other bacteria (Figure 5). However, a too-high dye concentration leads to its metabolites becoming dominant, which could block the active bacteria sites and become toxic to the bacteria and to the dye-degrading enzyme production process [41,42].

To develop cost-effective and sustainable solutions, agricultural waste biomasses such as rice husks and coconut carp are attractive options because of their low cost and high potential for pollutant absorption [11,43,44]. However, few studies have investigated the potential of rice husks as a significant carbon source for bacterial growth and as an accelerator in the dye degradation process. In a previous study, the azo dye Direct Red 75 was decolorized by 90% using a biofilter-sequenced anaerobic–aerobic system, with rice husks employed as carriers as a source of microorganisms [45]. In the current study, we introduced a strongly active extremophilic bacterial strain that has multiple functions—illustrating its high efficiency in dye degradation and using rice husks as a valuable carbon source for the bacteria’s growth and metabolism during the degradation process (Figure 6). The findings of this study could contribute to preventing secondary pollutants from being produced by rice husks that absorb dyes through surface absorption.

Few studies have investigated the bacterial strains that use raw lignocellulose agricultural waste as a carbon source to enhance dye degradation; a comparison of this study with previous studies is shown in Table 1.

Strain FW2 shows significant potential for various applications in the future because of its diverse functions. However, in-depth studies should be conducted on the molecular characteristics and potential metabolites produced by this bacterial strain in the future under certain operating conditions to determine a better scientific profile and control before large-scale application. Moreover, for further studies on the degradation of the components of the dyes, enzymes, and metabolites produced during dye degradation using strain FW2, other analyses should be performed: (1) sodium dodecyl sulfate–polyacrylamide gel electrophoresis (SDS-PAGE) for purifying and identifying proteins; (2) Fourier-Transform Infrared Analysis to recognize the degraded components of the dye and newly produced compounds; and (3) Gas Chromatography and Mass Spectrometry (GC and GC–MS) and (4) Nuclear magnetic resonance spectroscopy (NMR Spectra) to determine and identify the monomolecular structure of the degraded and produced compounds over the whole process of dye removal. Then, the possible mechanisms responsible for dye degradation can be proposed in a clear profile.

## 5. Conclusions

This study investigated the high potential for dye degradation in extremophiles. The multiple enzyme-producing *Bacillus* bacterial strain FW2 showed high degradation efficiency for MB and MO under heavy metal stress, alkaline conditions, and high salinity in both the absorption and accumulation pathways through the enzymatic pathways of LiP and Lac. In particular, rice husks played a significant role in accelerating the removal process and contributed to the efficient and robust degradation performance of dyes as a biosorbent and carbon source for bacterial growth—thereby preventing the release of secondary pollutants after the treatment process. Therefore, strain FW2—an extremophilic bacterium—is a promising candidate for large-scale application in dye degradation in the future, supported by cellulose-based agricultural waste such as rice husks as a valuable substrate. Further analyses on the molecular level should be carried out in the future to investigate the other useful applications of this bacterial strain in other fields of the pharmaceutical industry.

## Figures and Tables

**Figure 1 microorganisms-11-01974-f001:**
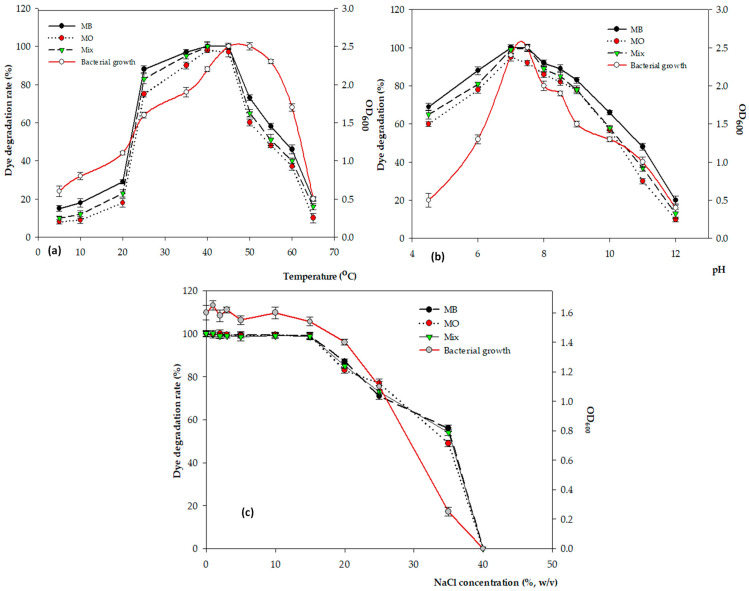
(**a**) Effect of pH, (**b**) temperature, and (**c**) NaCl on bacterial growth and dye degradation in the individual dyes and the mixture.

**Figure 2 microorganisms-11-01974-f002:**
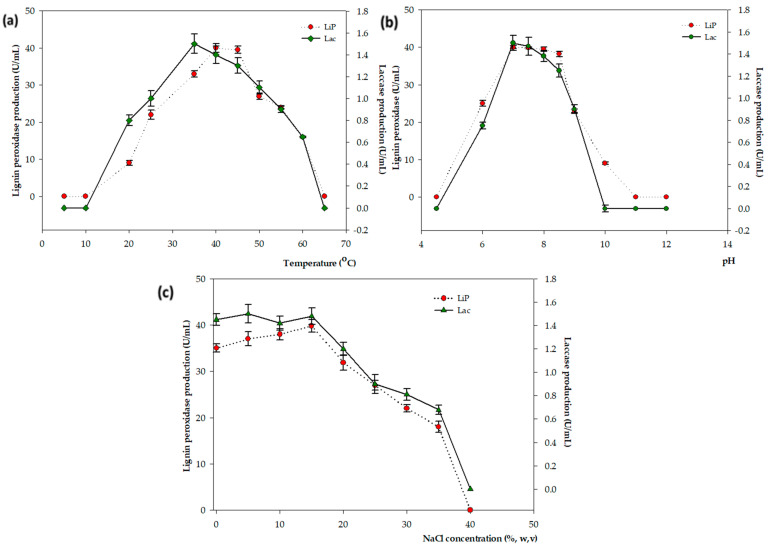
(**a**) Effect of pH, (**b**) temperature, and (**c**) NaCl on LiP and Lac production in strain FW2.

**Figure 3 microorganisms-11-01974-f003:**
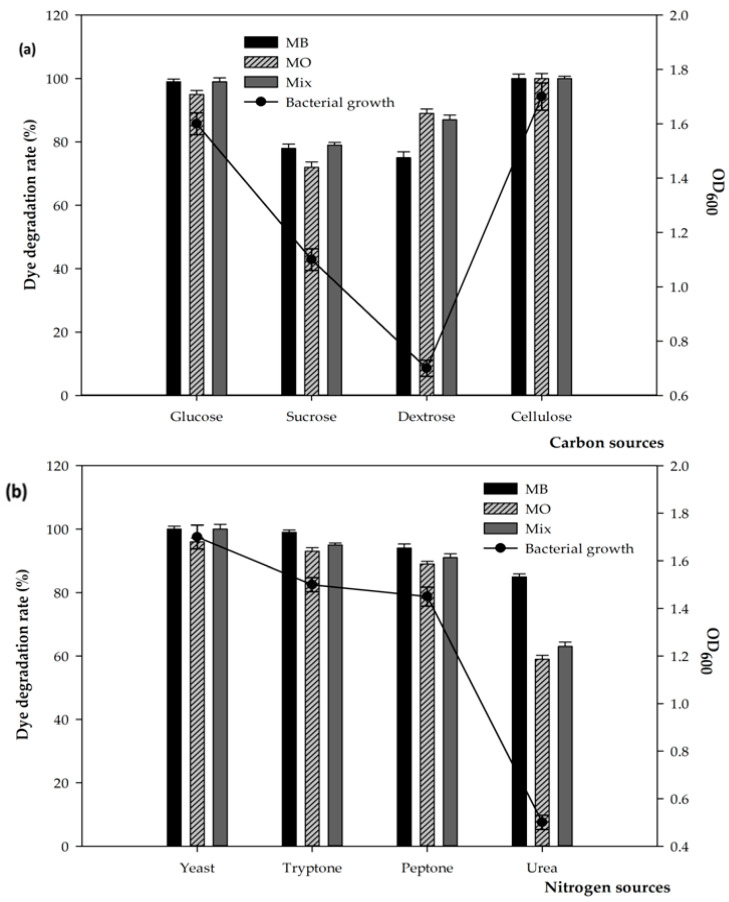
Effect of carbon (**a**) and nitrogen sources (**b**) on the removal of individual and mixed dyes.

**Figure 4 microorganisms-11-01974-f004:**
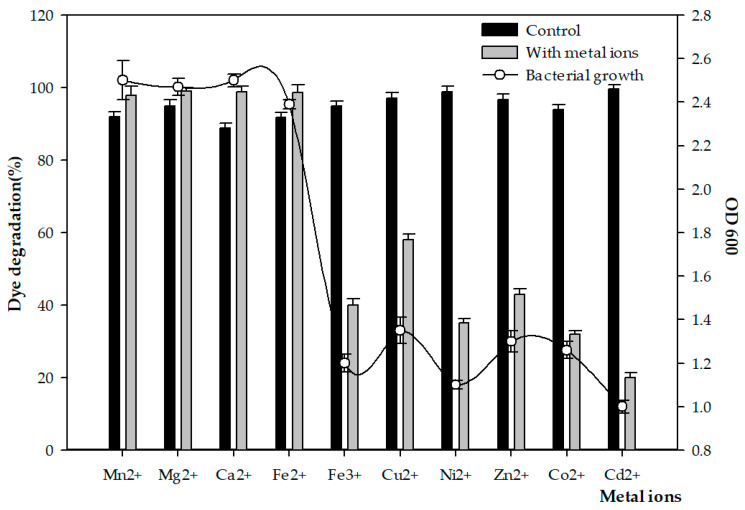
Mixed dye degradation by 10^8^ CFU/mL of FW2 inoculum in the presence of metal ions compared to control samples without metal ion addition. Experiment was carried out at 40 °C, 120 rpm for 5 days.

**Figure 5 microorganisms-11-01974-f005:**
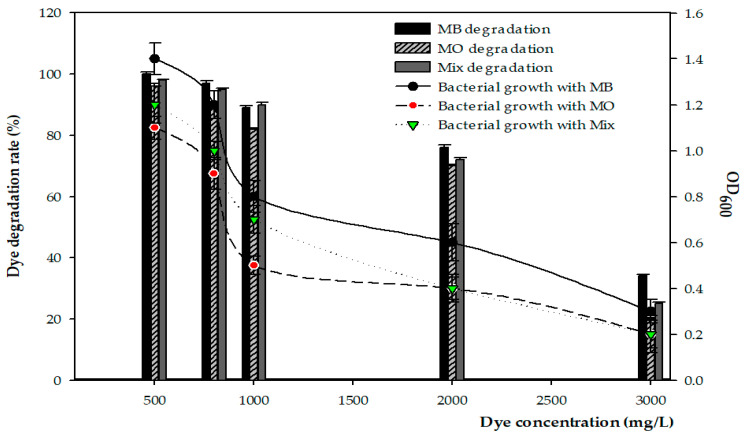
X-axis represents the different concentrations of dye that affected the degradation efficiency, using 10^8^ CFU/mL of FW2 inoculum, and the bacterial growth as measured at OD_600nm_ of FW2 was plotted on the right second *Y*-axis at 40 °C, 120 rpm.

**Figure 6 microorganisms-11-01974-f006:**
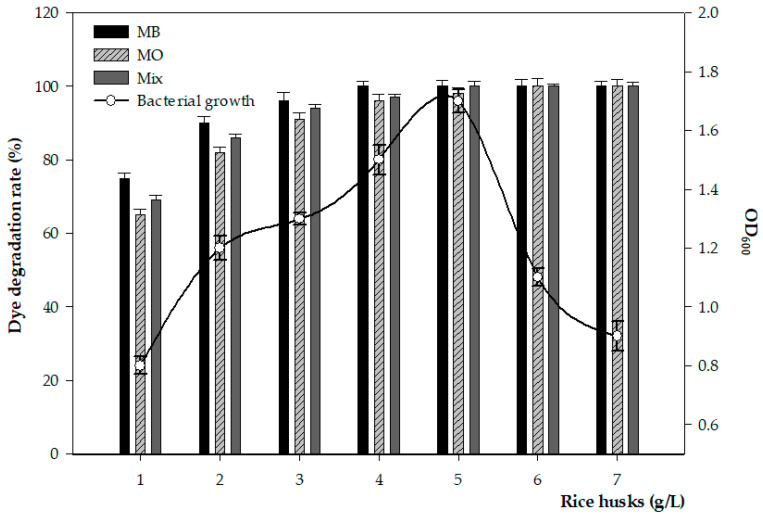
Rice husks’ role as a carbon source for bacterial growth that accelerates dye degradation process and enhance dye removal efficiency using 10^8^ CFU/mL of the FW2 inoculum.

**Table 1 microorganisms-11-01974-t001:** Summary of dye degradation of bacterial communities, co-cultures, and pure culture.

Degraded Dyes	Bacterial Strains	Carbon Source	Agricultural Waste	Efficiency	Degradation Mechanism	Reference
Reactive Black 5 and Reactive Red 2	Bacterial community	ND	Un-pretreated rice husks	80%	Enzymatic pathway	[46]
Crystalviolet, Congored, Methyleneblue and Safranin	Consortium	ND	Un-pretreated sawdust	Highest 77.2% MB degradation	Absorption	[47]
Reactive Black 5 and Reactive Red 152	Consortium of Halophilic bacterial strains	Glucose	ND	87% (Black 5), 85% (Red 152)	Enzymatic pathway	[16]
Methylene Blue	*Bacillus* React3	Tryptone	ND	99.50%	Enzymatic pathway	[11]
Methyl Orange	Bacterial community	ND	Unpretreated-rice husks	98%	Absorption	[19]
Methylene Blue	Consortium	ND	Pretreated-rice straw	49.6%	Enzymatic pathway	[48]
Methylene Blue and Methyl Orange	*Bacillus* FW2	Cellulose	Unpretreated-rice husks	100% (MO), 99.8% (MB), 98% (MB + MO)	Enzymatic pathways and accumulation	This study

## Data Availability

Not applicable.

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
