# Peer review of "Rice Husk—Cellulose-Based Agricultural Waste Enhances the Degradation of Synthetic Dyes Using Multiple Enzyme-Producing Extremophiles"

_microorganisms, 2023, doi:10.3390/microorganisms11081974_

Round 1

Reviewer 1 Report

Manuscript ID : microorganisms-2507307

My evaluation of the article led me to the conclusion that it raises some issues that should be addressed before publishing. The study's concept is sound and new to the field. I give revisions for the article's current format in order to bring it up.

General:

1.     The title and keywords are good.

2.     The abstract seems to be perfect and precisely stated. Please review the English errors in this section.

3.     What is novelty of the review?

4.     Author should highlight the research gap.

5.     Can you precisely highlight the novelty of the study, so the reader will not be confused?

6.     Page 1 Line 35 need a reference, please cite this one ``.Extraction of natural dyes from agro-industrial waste``.

7.     Page 1 Line 35 to 39 is a very long sentence which disturbs the meaning. Please revise your paper accordingly since some issue occurs on several spots in the paper.

8.     The whole introduction also needs a extensive English revision.

9.     Page 2 Line 48 need atleast two references. Please consider these references. (i) Role of nanomaterials in the treatment of wastewater: A review (2) Yaqoob, Asim Ali, et al. "Advanced Technologies for Wastewater Treatment." Green Chemistry for Sustainable Water Purification (2023): 179-202.

10.  The main objective of the work must be written on the more clear and more concise way at the end of introduction section.

11.  Page 1 Line 39 need another citation, please cite- Recent advances in metal decorated nanomaterials and their various biological applications: A review.

12.  Secion 2 is fine for me but there is some grammatical errors. Please check.

13.  Unfortunately, the quality of Figure 1 to 6 is very poor.

14.  Please provide space between number and units. Please revise your paper accordingly since some issue occurs on several spots in the paper.

15.  The results are simply demonstrated, but there is no comparison with the literature or scientific comment. Why?

16.  Please add a section about challenges and future perspectives.

17.   Please add a comparative profile section. The data is not enough so please add these two section to improve the standard to meet the journal standard.

18.  What about the replication of data?

19.  Conclusion section is missing some perspective related to the future research work, quantify main research findings, and highlight relevance of the work with respect to the field aspect.

20.  To avoid grammar and linguistic mistakes, MAJOR level English language should be thoroughly checked. Please revise your paper accordingly since several language issue occurs on several spots in the paper.

21.  Reference formatting need carefully revision. All must be consistent in one format. Please follow the journal guidelines.

Recommendation: Major REVISION

To avoid grammar and linguistic mistakes, MAJOR level English language should be thoroughly checked. Please revise your paper accordingly since several language issue occurs on several spots in the paper.

Author Response

Dear Reviewer,

We would like to thank you very much for your valuable time and for giving helpful comments to improve and evaluate my MS before being accepted. We responded all your comments. Please find it in the attachment. 

Best regards,

Van Pham

Reviewer 2 Report

This study investigated the degradation of synthetic dyes by multifunctional extremophilic bacterial candidates when the rice husks were used as carbon sources. This manuscript presented a systematic and comprehensive work. The content of the manuscript is in line with the research direction of Microorganisms. There are only a few minor mistakes with the content in the author's manuscript. The existence of these problems leads me to suggest minor revision.

(1)  The rice husks were used without any pretreatment and therefore may have inorganic or organic impurities. The basic analysis such as XRD and elemental analysis should be given.

(2)  Please check the label direction on the right axis of figures. It is difficult to read.

(3)  In Figure 4, x-axis labels are irregular.

(4)  In the “Results” section, the description of all figures should be more detailed.

Author Response

(The authors gave the same response as above.)

Round 2

Reviewer 1 Report

 Accept in present form